# Social Media Use and Depressive Symptoms—A Longitudinal Study from Early to Late Adolescence

**DOI:** 10.3390/ijerph17165921

**Published:** 2020-08-14

**Authors:** Kati Puukko, Lauri Hietajärvi, Erika Maksniemi, Kimmo Alho, Katariina Salmela-Aro

**Affiliations:** 1Faculty of Educational Sciences, University of Helsinki, 00014 Helsinki, Finland; lauri.hietajarvi@helsinki.fi (L.H.); erika.maksniemi@helsinki.fi (E.M.); katariina.salmela-aro@helsinki.fi (K.S.-A.); 2Department of Psychology and Logopedics, Faculty of Medicine, University of Helsinki, 00014 Helsinki, Finland; kimmo.alho@helsinki.fi

**Keywords:** social media, depressive symptoms, adolescence, longitudinal study, cross-lagged panel model

## Abstract

An increasing number of studies have addressed how adolescents’ social media use is associated with depressive symptoms. However, few studies have examined whether these links occur longitudinally across adolescence when examined at the individual level of development. This study investigated the within-person effects between active social media use and depressive symptoms using a five-wave longitudinal dataset gathered from 2891 Finnish adolescents (42.7% male, age range 13–19 years). Sensitivity analysis was conducted, adjusting for gender and family financial status. The results indicate that depressive symptoms predicted small increases in active social media use during both early and late adolescence, whereas no evidence of the reverse relationship was found. Yet, the associations were very small, statistically weak, and somewhat inconsistent over time. The results provide support for the growing notion that the previously reported direct links between social media use and depressive symptoms might be exaggerated. Based on these findings, we suggest that the impact of social media on adolescents’ well-being should be approached through methodological assumptions that focus on individual-level development.

## 1. Introduction

Social networking services (SNSs), such as Facebook, Instagram, and Snapchat, have become intertwined with adolescents’ daily lives. According to the Pew Research survey, 89% of US teens report their social media use to range from constant to several times a day [1]. In Finland, 15–16-year-olds spend on average 4 h 22 min daily online [2]. This high degree of connectivity has been accompanied by concerns that the time that adolescents use on social media can negatively influence their psychosocial well-being, putting them at risk for depressive symptoms [3,4], especially since recent statistics indicate a rapid and unprecedented increase in reported rates of depressive symptoms in this age group [5,6,7]. Nevertheless, there is little agreement on whether social media has a role in these increases—it could either be a cause or a consequence, or causally unrelated.

This study examined longitudinal within-person associations between the frequency of active social media use and depressive symptoms from early to late adolescence. We use the term “early adolescence” for ages 13–14, “middle adolescence” for ages 15–17, and “late adolescence” for ages 18–19. We employed social media use as an umbrella term covering internet-based networking activities that enable adolescents to interact with others, verbally and visually [8]. We focused particularly on active social media use, which refers to socially-oriented SNS use, such as sending messages, sharing updates, and liking other people’s doings on SNSs [9]. These activities are different from passive social media use, which refers to SNS-related activities without a strong direct connection, such as browsing newsfeeds or other people’s profiles [9]. Additionally, we focused on depressive symptoms, which are one of the most common disabling mental health issues in adolescence [10]. Depressive symptoms cover irritability, depressed mood, loss of interest and pleasure in activities, loss of confidence, and sleeping difficulties [11].

We intended to contribute to the field by using a six-year longitudinal research design and analyzing the associations at the individual level of development. Specifically, we examined the within-person associations between social media use and depressive symptoms. With this approach, our aim was to extend knowledge about changes between adolescents’ social media use and depressive symptoms over time. The results are discussed within the theoretical framework of displacement approach [12] and social compensation approach [13,14]. Adolescence is a critical developmental period for the onset of serious mental health problems [15], and depressive symptoms are associated with substantial negative health effects both in adolescence and later in life [16]. Therefore, it is crucial to study risk factors for depressive symptoms in adolescence and gain a deeper understanding of the potential impact of social media.

### 1.1. Associations between Social Media Use and Depressive Symptoms

There are two primary theoretical approaches explaining how and why social media use and depressive symptoms might be related. The first, the displacement approach, contends that time spent in online environments occurs at the expense of alternate activities linked to psychological well-being, such as exercise and spending time with family and friends [12]. The theory suggests that the increases in psychosocial problems are driven by changes in the way adolescents interact with each other through social media and the time they spend online. From this perspective, the time that adolescents spend on social media might be related to subsequent depressive symptoms. The second, the social compensation approach, suggests that adolescents with pre-existing mental health difficulties may turn to social media to make up for real or perceived psychosocial problems [13,14]. Adolescents who are depressed tend to experience emotional distress and interpersonal difficulties [17,18]. In this case, an adolescent who is feeling depressed might turn to social media as a form of escapism to relieve this distress and connect with others online.

However, drawing conclusions about the potential association from the existing research is premature. Systematic reviews and meta-analyses on adolescents’ social media use and psychological well-being have produced mixed results [9,19]. Several reviews of social media use and its associations with depressive symptoms have reported small positive correlations [20,21,22,23]. For instance, in a systematic review of 11 studies of children and adolescents, McCrae, Gettings, and Pursell [23] concluded that the mean correlation between social media use and depressive symptoms was positive and low (mean r = 0.13, 95% confidence interval, CI [−0.05, 0.20]). Another review examining the association between frequency of time spent on SNSs and depression noted that eight studies reported a direct positive association with depression, while almost twice as many found no significant associations [24], indicating that frequency of SNS use was only weakly associated with depressive symptoms. Most reviews and meta-analyses have not provided strong support for a robust link between social media and depressive symptoms, instead reporting correlations within individual studies to be on average small and heterogeneous [25,26,27].

A few longitudinal studies have been conducted on the associations between social media and depressive symptoms. Some of these studies have suggested a possible causal link from social media use to higher levels of depressive symptoms over time [28,29]. For instance, an influential study of two nationally representative samples of U.S. adolescents found a positive correlation from social media use to later depressive symptoms [29]. The authors concluded that social-media use may be driving increases in depression among U.S. adolescents. However, longitudinal evidence regarding the direction of effects remains inconsistent. Other researchers have found a directional relationship from depressive symptoms to subsequent social media use [30] and some studies have reported bi-directional effects, suggesting that depressive symptoms can be both a result and a cause of the time adolescents spend on social media [31,32]. Yet, some longitudinal studies have not found significant associations [33,34]. For instance, in a longitudinal study, Nesi and colleagues [33] observed that the frequency of social media use and adolescent depressive symptoms were unrelated. However, they tested only one direction of the possible effect, from depressive symptoms to social media use. Thus, it remains unclear whether the association between social media use and depressive symptoms is causal, bi-directional, or non-existent. Additionally, as most of these longitudinal studies have been of very short duration, ranging from just few months to two years, the existing associations between social media use and depressive symptoms have not been investigated in depth across adolescence.

Additionally, most of the existing longitudinal studies and meta-analyses have detected associations between social media use and depressive symptoms only at a between-person level by comparing differences between individuals at the same time point. These between-person associations do not provide information about within-person processes. At a within-person level, it is possible to track individuals’ personal changes over time. [35] In other words, it is possible to compare an individual’s level of depression and social media use across different time points. In the context of social media effects research, differentiating these levels might be crucial, as between-person effects may substantially differ from within-person effects. This may lead to different conclusions based on the same data [35]. For instance, in a recent longitudinal study, Coyne and colleagues [36] found that time spent using SNSs was positively related to depression on the between-person level over eight years in adolescence. In contrast, at the within-person level, these effects disappeared. Although social media scholars have begun to voice concern over using between-person analyses to make assumptions about social media effects on adolescents’ well-being [37], the direction or the stability of the associations between social media use and depressive symptoms is rarely examined at the within-person level.

Moreover, the associations between social media use and adolescents’ mental health might be more nuanced and dependent on several individual differences between adolescents. For instance, females tend to use social media to connect with existing social networks, while males use social media for shared computer-based activities, such as online gaming [38]. Moreover, adolescents’ access to social media are bound up with structural inequalities in family’s socioeconomic status with adolescents from wealthier families having better digital skills and access to SNSs than their peers with lower socioeconomic status [39]. Regarding depressive symptoms, females usually experience higher levels of depressive symptoms compared to males from approximately age 13 onwards [40], and adolescents with low socioeconomic status are likely to experience greater depressive symptoms across adolescence [41]. These variables might modify the strength of the effect of social media use on a given outcome.

### 1.2. Aims

The associations between social media use and depressive symptoms have been under scientific scrutiny for some time now. However, the mixed results, methodological limitations, and heavy reliance on cross-sectional research designs make it difficult to determine whether social media use is linked to depressive symptoms across adolescence in any way. Additionally, the directions of effects are rarely examined with research methods focusing on the individual-level development. To fill these research gaps, the current study examined within-person effects between social media use and depressive symptoms over a six-year period in adolescence. Specifically, we tested both directions of the postulated associations between social media use and depressive symptoms. We also tested whether any associations remain after adjusting for the potential influence of participant’s gender and the family financial status

## 2. Materials and Methods

### 2.1. Data Collection and Participants

The present study was a large-scale longitudinal cohort study targeting adolescents born in the year 2000. The data collection was conducted over a six-year period from 2014 (T1) to 2019 (T6) with one-year lags. In 2017 (T4), however, no data were collected. The sample contained a total of 2891 adolescents (42.7% male) living in the capital area of Finland. The participants were 13–14 years old when the study started and 18–19 years old when it ended. Additionally, the study targeted new participants each year. Out of the total 2891 participants 684 (23.6%) participated three or more times over the data collection period, 815 (28.2%) two times, and 1145 (39.6%) only once.

At each of the five data collection points, the participants fill out a self-report questionnaire on their social media use, psychosocial well-being, and various background variables. During 2014–2018 the data collection was conducted in collaboration with schools. This meant that all schools that were able to organize the data collection administered the questionnaire in classroom settings, and all students who were present in the class and willing to complete the questionnaire were included as participants. Participation in the study was voluntary and informed consent forms were collected from the students and from their guardians at the beginning of the study and again in 2018, when the participants were in the second year of high school. In 2018, the participants who were not reached through schools were asked to fill out the questionnaire using text messages. In 2019, the data collection was organized only by contacting adolescents via text messages. In 2018 and 2019, all participants who filled out the questionnaire were rewarded with a gift card valued at ten euros. The study protocol was approved by the University of Helsinki Ethics Review Board for the Humanities and Social and Behavioural Sciences.

### 2.2. Measures

#### 2.2.1. Depressive Symptoms

Depressive symptoms were assessed at all five measurement times by the Finnish version of the Depression Scale [42]. The DEPS scale has been a popular self-rating questionnaire for screening depressive symptoms in the general population and for identifying high-risk groups in Finland and in many neighboring countries [43,44,45]. The scale consists of 10 items that are answered on a 4-point Likert-type scale, ranging from 1 (‘not at all’) to 4 (“very much”). An example item is “The future is hopeless.” In the current study, the scale showed good internal consistency, with the Cronbach alpha ranging between 0.92 and 0.94 across the measurement times (see Table 1).

#### 2.2.2. Social Media Networking

Frequency of active social media use was measured using the social media networking dimension of the socio-digital participation inventory [46]. The validity of the scale has been established across different studies in Finland with different age groups [47,48,49]. The scale consists of four items measuring frequency of active, socially oriented use of SNS such as chatting, sharing photos and status updates, or posting personal content on social networking sites. The items were as follows: “I follow my friends’ profiles, pictures, and updates”, “I update my status and share content with others”, “I chat (e.g., Whatsapp, Facebook, e-mail)” and “I share pictures and picture updates of my doings taken with my phone (e.g., Instagram)”. The items were rated on a seven-point Likert-type frequency scale ranging from 1 (“never”) to 7 (“all the time”). The Cronbach alpha reliabilities for the sum scores were between 0.71 and 0.84 across the measurement times (see Table 1).

#### 2.2.3. Covariates

Gender and family financial status were included as covariates in sensitivity analyses. Family financial status was measured at all five time points with one item enquiring about general financial status of the family: “How would you evaluate your family’s financial situation?” on a scale from 1 (”bad”) to 5 (“good”); with one item on family financial status compared with peers: “How would you evaluate your family’s economic situation compared with your friends’ families?” with the response options: 1 (“richer”), 2 (“poorer”), 3 (“the same”) and 4 (“I don’t know”) and with one item enquiring about own money: “How much money do you have personally?” on a scale from 1 (“little”) to 5 (“a lot”). These three items were combined into one general variable measuring family financial status situation that was rated on a three-point scale from 1 (“bad”) to 3 (“good”). The participants were also asked to report their gender, coded as 0 (“female”) or 1 (“male”). In total 81 participants chose the option “other” or did not report their gender at all. Those respondents were coded as missing values in the data.

#### 2.2.4. Analysis Strategy

To investigate the longitudinal within-person effects between adolescents’ social media use and depressive symptoms, we applied a Random Intercept Cross-lagged Panel Model approach (RI-CLPM) [38]. The specific feature of RI-CLPM is that it distinguishes the variance of observed scores into variance of between-person and within-person fluctuations over time [35]. In the context of the present study, the RI-CLPM allowed us to specify whether elevated social media use would be associated with subsequent changes in depressive symptoms over time, or vice versa. More specifically, the model controls for time-invariant trait-like individual differences in social media use and depressive symptoms, such that more insight is provided into how these two constructs are linked at an intra-individual level [35].

For the purpose of this study, we constructed three RI-CLPM models. First, we estimated the relationship between social media use and depressive symptoms across five measurement points. Second, after confirming the model, we added gender and family financial status as covariates to examine whether the associations between depressive symptoms and social media use are robust. Third, we conducted some sensitivity test with a smaller subsample of participants. In these models, four types of effects are provided: between-person correlations, within-time associations (correlated change), within-person stability effects (autoregressive paths), and within-person cross-lagged effects. To capture the time-invariant differences between persons in depressive symptoms and social media use, we included two random intercept factors for each measure. The two random intercept factors indicate the between-person variances (i.e., stable time-invariant trait aspects) of social media use and depressive symptoms over time. The within-person processes were represented by autoregressive and crossed paths between the latent fluctuations of social media use and the latent fluctuations of depressive symptoms across time around the participants’ own means (see Figure 1).

The model fits were evaluated using Chi-square values and the root mean square error of approximation (RMSEA) with an approximate acceptable cut-off value of less than 0.08, standardized root mean residual (SRMR) with an approximate cut-off of less than 0.08, and incremental indices, such as comparative fit index (CFI) and Tucker-Lewis Index (TLI), with approximate cut-off values of more than 0.09 [50]. Correlation analyses and RI-CLPM were conducted using Mplus 8.0 [51] in conjunction with R and R Studio [52,53] with the package MplusAutomation (Muthén & Muthén, Los Angeles, CA, USA) [54]. The model parameters were estimated using the MLR estimator because the depression scores were slightly skewed (see Table 1). Social media use, depressive symptoms, and family financial status were used as sum scores in the models. The syntaxes and research data can be found following this OSF identifier: doi:10.17605/OSF.IO/WPF8Z.

## 3. Results

### 3.1. Missing Values

As a preliminary analysis, the data were screened for the number and patterns of missing values using the IBM Statistical Package for Social Sciences (SPSS; version 25, SPSS Inc., Chicago, IL, USA) at each time point separately to evaluate the sparsity and quality of the questionnaire data. The items had less than 6.3% missing values at the first measurement time (T1). Little’s MCAR test showed that the missing values were not missing completely at random (χ^2^(1156) = 1376.723, *p* = 0.000). However, the normed Chi-square test (χ^2^/df) suggested only small violations of the MCAR assumption (1376.723/1156 = 1.1909). At the second measurement time (T2), Little’s MCAR test revealed that data were missing completely at random (χ^2^(550) = 595.814, *p* = 0.086), and there was less than 12.0% missing values in all items. At the third measurement time (T3), items measuring social media use had 2.3% missing values, items measuring depressive symptoms 21.7% missing values, and items measuring family’s economic background 19.0% missing values. Little’s MCAR test showed that the missing values were not missing completely at random (χ^2^(424) = 580.441, *p* < 0.001), but the normed Chi-square test was acceptable (580.441/424 = 1.3689). At the fifth measurement time (T5), items measuring social media use and depressive symptom had less than 6.8% missing values, and items measuring family’s financial status had 11.3% missing values. Little’s MCAR test was significant (χ^2^(455) = 612.816, *p* < 0.001), but again the normed Chi-square test implied only small violations (612.816/455 = 1.3468). At the last measurement time (T6), the MCAR assumption held (χ^2^(145) = 145,354, *p* = 0.476), and there were less than 2.9% missing values in all questionnaire items.

The missing data were handled using full information maximum likelihood (FIML) method. FIML method uses all available data in order to estimate the model without imputing data [55]. The method has been shown to work well in reducing bias in longitudinal studies even with systematic attrition and perform better than deletion based methods even with very high rates of missing data [56,57]. Consequently, we were able to estimate the models using the full sample of 2891 participants who supplied information at least at one measurement point without relying on the suboptimal deletion of participants. However, due to the substantial amount of non-monotone missing data, sensitivity checks were carried restricting the analysis to a subsample of 684 participants who had answered the questionnaire at least three times.

### 3.2. Descriptive Statistics and Internal Consistencies of the Scales

The descriptive statistics and internal consistencies among the key study variables are presented in Table 1. At all measurement times, social media use was common among respondents, whereas the average levels of depressive symptoms were low. The means and standard deviations increased slightly between the first (T1) and fifth (T5) measurement times, suggesting a small increase at the group level in terms of both depressive symptoms and social media use.

### 3.3. Bivariate Correlations

We conducted a preliminary analysis to test the bivariate correlations between each time point. The bivariate correlations revealed some small positive associations between depressive symptoms and social media use between T2 and T3 and between T3 and T5 (correlation coefficients varied between r = 0.10 and 0.15). However, most of these correlations were weak or non-existent. Females suffered more from depressive symptoms and used social media more frequently than males. For more details, see Table 2.

### 3.4. Within-Person Effects between Social Media Use and Depressive Symptoms

Table 3 presents the model fit indices for all of the estimated RI-CLPMs. All models fitted the data well (according to the criteria of RMSEA < 0.08, CFI > 0.90, and TLI > 0.90). Based on the model fit indices, the autoregressive and cross-lagged paths were constrained to be equal across time in all models. Based on the modification indices for Model a, in Model b the autoregressive path between T5 and T6 on depressive symptoms was set free. This might be because the carry-over effect for depressive symptoms is stronger in late adolescence or an artefact of the sampling design. In Model c, gender and family financial status were included as covariates.

The results in Model b revealed a small, positive within-person effect from depressive symptoms to social media use a year later (B = 0.13, *p* = 0.039, 95% CI [0.01, 0.25]). This indicates that an individual’s higher than own average depressive symptoms predicted a slight increase in social media use a year later. The reverse within-person effect from social media use to later depressive symptoms was not significant (B = 0.01, *p* = 0.802, 95% CI [−0.05, 0.07]). This shows that adolescents who reported higher levels of depressive symptoms than their typical levels did not report more or less social media use at a later time. The estimates for the within-person correlated change between social media use and depressive symptoms were approximately zero (B = 0.02, *p* = 0.233, 95% CI [−0.01, 0.04]), meaning that when students reported higher social media use above their mean they did not simultaneously report greater depressive symptoms than their mean level. Statistically significant parameter estimates were not found between T3 and T5 (the two-year lag). The parameter estimates for Model b are presented in Table 4 and standardized within-person effects (β) in Figure 2.

### 3.5. Results of Sensitivity Analyses

The results for sensitivity analyses are presented in Table 5 and Figure 3. The magnitude of the effects between social media use and depressive symptoms remained largely similar after adjusting for the influence of participant’s gender and the family financial status. However, the small positive effect from depressive symptoms to later social media use became non-significant when considered in the light of the conventional criteria of *p* < 0.05 (B  =  0.12, *p* = 0.060, 95% CI [−0.01, 0.23]). Additionally, the parameter estimates remained largely unchanged when the analysis waslimitedonly to those who had participated the study at least three times. For more details, see Table A1 and Table A2.

## 4. Discussion

This study extends earlier research on the associations between active social media use and depressive symptoms by examining the longitudinal within-person effects in early and late adolescence. A considerable amount of research has been dedicated to the understanding of these associations. However, most of these studies have relied on cross-sectional research designs and detected associations between social media use and depressive symptoms only at the between-person level. Therefore, it is unclear whether depressive symptoms are a cause or a consequence of social media use. We also tested whether the associations remain after controlling for the potential influence of participant’s gender and the family financial status.

Our results provide little evidence that active, socially-oriented social media use and depressive symptoms develop together in adolescence at the within-person level. The average levels of social media use and depressive symptoms grew steadily across adolescence, but the within-person associations between adolescents’ social media use and depressive symptoms were very small and somewhat inconsistent over time. The effects remained largely unchanged when gender and family’s financial status were controlled. Additionally, most of the bivariate correlations were weak or non-existent. Thus, these results contradicts previous claims that social media use leads to greater depressive symptoms [28,29] and provide support for the growing notion that the previously reported associations between social media use and depressive symptoms might be exaggerated [25,26,27]. One explanation for this is that many of the studies supporting displacement effects come from a time when only a minority of young people were socializing online [9,19]. Today, youth increasingly turn to social media as a primary means of interaction with peers [58]. As frequent social media use has become the norm in adolescents’ daily lives, it may be that the previously reported effects of ‘screen time’ have dissipated over time. The present results also fit well with the developmental view of depression as a multifactorial process involving various social, individual, and situational factors [16]. Therefore, it is unlikely that the frequency of social media use reveals the whole truth.

Furthermore, the results demonstrate that depressive symptoms predicted slight increases in active social media use at the within-person level across adolescence. In line with the social compensation approach, this finding could indicate that adolescents’ social media use increases when they experience depressive symptoms [13,14]. Given that adolescents with depression experience interpersonal difficulties [17,18], individuals with underlying depression may be more drawn to social media interactions than to face-to-face interactions. However, there are concerns that spending time online might lead psychologically vulnerable adolescents to reach pathological levels of internet or social media use, as they are trying to compensate for problems in their offline lives [59,60]. Nevertheless, adolescents may not see their social media use as a problem, but as a way to respond to psychosocial problems in their lives. An emerging body of research suggests that online environments can be used in ways that support adolescent mental health [61,62,63]. For instance, a recent nationally representative survey in the US on online help-seeking behavior indicated that a significant number of teens and young adults who experienced moderate to severe symptoms of depression reported turning to the internet for help and social support [63]. In this sense, the small within-person correlation from depressive symptoms to later social media use found in the present study may be interesting. The results highlight the need for deeper discussion about the outcomes and potential benefits of social media, especially for vulnerable adolescents. It would also be worthwhile to address whether social media can provide a tool to detect mental health symptoms, for instance through analysis of the content created by users.

From a theoretical point of view, the absence of significant within-person effects indicates a need to clarify existing theoretical hypotheses on the links between social media use and depressive symptoms. Today, most of the studies have focused on the quantity of social media use based on theories such as the displacement approach [12]. However, our findings suggest that the associations appear to be more reflective of pre-existing levels of depressive symptoms of the individual than direct displacement effects of screen time. In the future, scholars should make more explicit theoretical hypotheses regarding individuals and consider various other factors that may modify the direction or the strength of the effect of social media use on a given outcome. For instance, fear of missing out [64] and perceived social support in online interactions [65] might be more influential than the quantity of social media use. Given that there are novel methodological approaches to investigate these nuances in social media effects, such as the Random Intercept Cross-Lagged Panel Model [36] used in this study, these methodological approaches may help advance the field.

### Strengths and Limitations

This study has some strengths and limitations that are worth considering. The recent data collection, longitudinal design, and large sample size strengthen the developmental inferences that can be drawn from the analyses. Additionally, the Random Intercept Cross-Lagged Panel Model, in which covariance was divided into between-person and within-person effects, provided a more accurate understanding of how social media use and depressive symptoms are linked over time at the individual level. Both the social media networking and depressive symptoms scales used in this study demonstrated good psychometric properties in terms of levels of scale-score alpha reliability (Table 1). Furthermore, our measure of social-media networking captured various socially driven, active, practices on SNS rather than average screen time [21,22,23,24].

As a limitation, it needs to be noted that the sample was not very representative, but instead included only adolescents living in the Helsinki metropolitan area with a somewhat homogeneous socio-economic status. Therefore, this work should be replicated using a representative sample. Additionally, the measurement gap of two years between T3 and T5 may have further reduced the initially weak correlations found in this study. In addition, there were a substantial amount of non-monotone missing data. Given that schools were responsible for conducting the data collection during 2014 and 2018, the attrition could have related to either the schools’ or the teachers’ inability to organize the collection that year or students being absent at the time of data collection. We, however, chose to include all available data using full information maximum likelihood method (FIML), which in general was expected to provide us estimates with less bias than with data deletion based methods [55,56,57]. In addition, the sensitivity analyses we conducted with a more conservative approach to missing data did not change the main conclusions of the present study. However, it needs to be acknowledged that the large amount of missing data increases sampling variation and the need to rely on model-based assumptions, which is not optimal. Therefore, the inferences are drawn from the most parsimonious model, in which the structural parameters were fixed to equality across similar time intervals that fit the data well. Still, the estimates reported in the present study are strongly model-based and should be treated as such.

Another concern is that family financial status and gender were only controlled in the sensitivity analysis. The effect of social media on adolescents’ well-being is increasingly understood as a complex interplay between various individual, social and contextual factors [66]. That is, some adolescents might be more susceptible to social media effects than others. For instance, digital practices in families and parental mediation styles appear to make a difference in the ways adolescents engage in social media [67]. Previous studies suggest that social media effects might be dependent individual characteristics, such as personality traits, shyness and attachment style [68,69]. Other studies have highlighted the mediating role of social comparisons and perceived social support in online interactions [70,71]. Overall, the multifactorial nature of depressive symptoms in adolescence makes it difficult to discuss these associations without acknowledging various biological, interpersonal, cognitive and social determinants. In this study, due to the available data, we were not able to take into account these factors inclusively. Clearly, more research is needed to understand these nuances and susceptibilities in social media effects.

In the present study, we focused particularly on frequency of active social media use, but did not investigate qualitative differences in social media use. A number of studies suggest that social media can have differential effects upon wellbeing depending whether adolescents engage in specific online activities, such as messaging with friends and family, browsing social networking sites or posting updates [22,48,59]. For instance, in a representative sample of Icelandic adolescents, passive social media use (i.e., scrolling and hanging out) increased depressive mood while active social media use (i.e., using social media to communicate) decreased it [72]. Furthermore, information on adolescents’ social media use was based on retrospective self-report questionnaires, which may be biased [73]. Research suggest that self-report measures of screen time only moderately reflect the actual use as most individuals usually underestimate their media use [74]. Therefore, we do not assume that different measures that would account these differences in social media use would yield the same results. Since social media and the way that adolescents use it are changing constantly, the measures to be used in future studies should be more precise, ideally objective. Future studies could examine, for instance, how these longitudinal associations translate into short-term effects with data derived via experience sampling method or ecological momentary assessment [75,76].

## 5. Conclusions

Much of the public debate about the role of social media in adolescents’ mental well-being focuses on the potential negative effects. However, our findings contradict the popular narrative that social media use leads to increased depressive symptoms among adolescents. The present results provide some small support for the view that adolescents’ social media use might increase when they experience greater depressive symptoms. Yet, this effect was very small, statistically weak, and somewhat inconsistent over time. We hope that these results help to minimize the risk of informing public policy with misleading information and allow the discussion to move beyond the displacement debate. A future venue might be to study social media effects through methodological and theoretical approaches that emphasize individual development.

## Figures and Tables

**Figure 1 ijerph-17-05921-f001:**
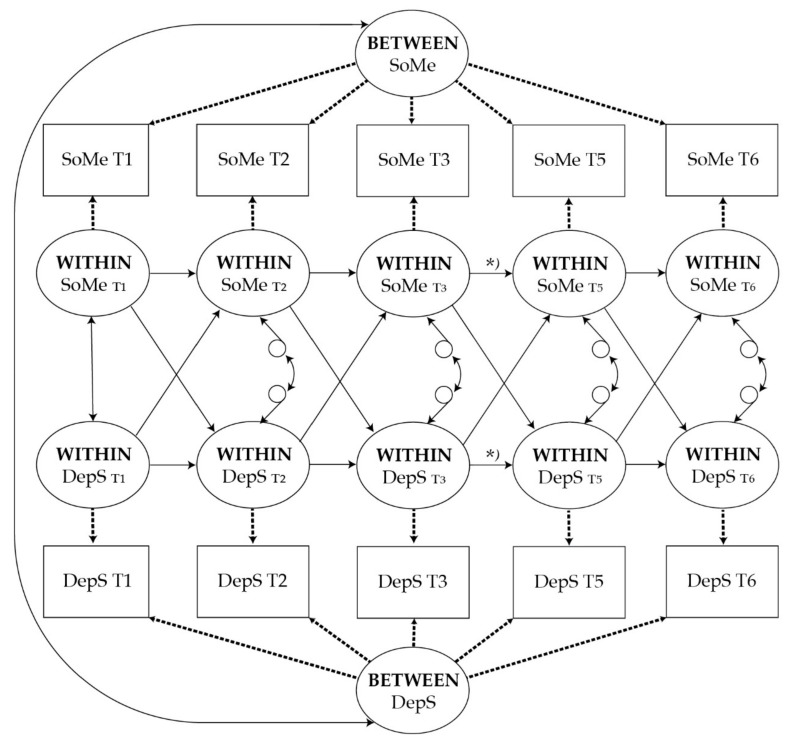
Random Intercept Cross-Lagged Panel Model Linking Social Media use (SoMe) with Depressive Symptoms (DepS) from early to late adolescence. T1–T6 indicate the time points within the data collection. Dashed lines indicate paths that were fixed to one. Solid lines between the within-level variables indicate that the paths were set equal across time (excluding paths T3–T5). Note: *, two-year gap in data collection.

**Figure 2 ijerph-17-05921-f002:**
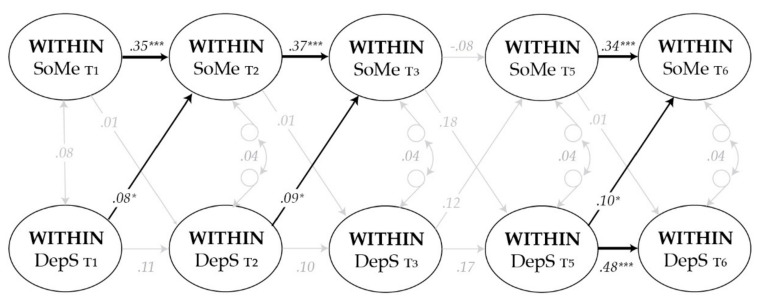
Standardized within-person effects (β) between social media use (SoMe) and depressive symptoms (DepS). Thicker black lines represent significant effects (*** *p* < 0.001), narrower almost significant effects (* *p* < 0.05), grey lines non-significant effects (*p* > 0.05). Correlated changes are presented with two-way arrows. Autoregressive and cross-lagged paths were set equal across time, excluding paths between T3 and T5 (standardized estimates differ slightly due to different variances).

**Figure 3 ijerph-17-05921-f003:**
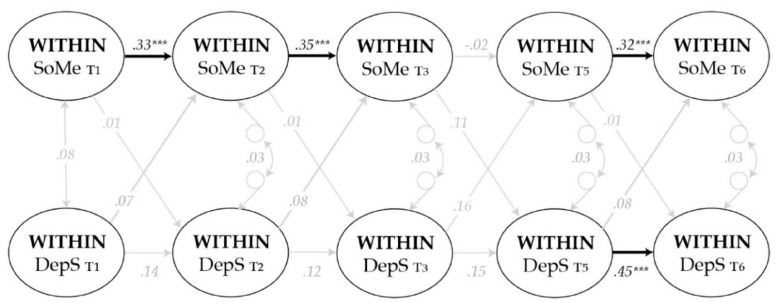
Standardized within-person effects (β) between Social Media use (SoMe) and Depressive Symptoms (DepS) when gender and family’s financial status are included as covariates. Thicker black lines indicate significant effects (*** *p* < 0.001) and grey lines indicate non-significant effects (*p* > 0.05). Correlational changes are presented with two-way arrows. Autoregressive and cross-lagged paths were set equal across time, excluding paths between T3 and T5 (standardized estimates differ slightly due to different variances).

**Table 1 ijerph-17-05921-t001:** Summary of descriptive statistics and internal consistencies.

	N	M	SD	SE	Min	Max	Skew	Kurtosis	*α*
T1									
Social Media Use	1302	4.02	1.11	0.03	1	7	−0.41	−0.14	0.84
Depressive Symptoms	1272	1.57	0.62	0.02	1	4	1.56	2.27	0.92
T2									
Social Media Use	1169	4.03	1.09	0.03	1	7	−0.37	−0.33	0.81
Depressive Symptoms	1077	1.58	0.63	.02	1	4	1.39	1.61	0.93
T3									
Social Media Use	930	4.20	1.05	0.03	1	7	−0.39	0.05	0.78
Depressive Symptoms	1433	1.71	0.76	0.02	1	4	1.03	0.44	0.94
T5									
Social Media Use	1032	4.27	1.03	0.03	1	7	−0.14	0.11	0.71
Depressive Symptoms	1040	1.81	0.69	0.02	1	4	0.94	0.38	0.93
T6									
Social Media Use	590	4.24	0.96	0.04	1	7	−0.00	0.52	0.71
Depressive Symptoms	582	1.98	0.76	0.03	1	4	0.75	−0.33	0.93

Note: T4 is missing because in 2017 no data were collected.

**Table 2 ijerph-17-05921-t002:** Bivariate correlations between each measurement point (N = 2891).

	1.	2.	3.	4.	5.	6.	7.	8.	9.	10.	11.	12.	13.	14.	15.	16.
1. Gender	1															
2. SoMe1	−0.28	1														
3. SoMe2	−0.31	0.65	1													
4. SoMe3	−0.31	0.57	0.70	1												
5. SoMe5	−0.20	0.39	0.49	0.52	1											
6. SoMe6	−0.19	0.38	0.49	0.55	0.64	1										
7. DepS1	−0.15	0.07	0.09	0.05	0.09	0.11	1									
8. DepS2	−0.19	0.03	0.10	0.15	0.09	0.11	0.47	1								
9. DepS3	−0.09	−0.02	0.04	0.04	0.12	0.05	0.37	0.49	1							
10. DepS5	−0.20	0.08	0.03	0.09	0.04	0.03	0.39	0.52	0.45	1						
11. DepS6	−0.19	0.03	0.08	−0.01	0.09	0.07	0.35	0.36	0.46	0.65	1					
12. FinS1	0.05	0.02	0.00	−0.02	−0.01	−0.07	−0.21	−0.13	−0.08	−0.10	−0.07	1				
13. FinS2	0.07	0.07	0.02	0.00	−0.05	0.05	−0.11	−0.21	−0.10	−0.17	−0.04	0.46	1			
14. FinS3	0.10	−0.01	0.05	0.00	0.12	0.05	−0.14	−0.14	−0.10	−0.20	−0.11	0.45	0.46	1		
15. FinS5	0.10	0.15	0.03	−0.02	0.03	0.01	−0.20	−0.17	−0.18	−0.24	−0.20	0.47	0.37	0.52	1	
16. FinS6	0.05	0.03	−0.07	−0.14	−0.04	−0.03	−0.24	−0.14	−0.22	−0.19	−0.22	0.37	0.33	0.43	0.58	1

Note: SoMe: Social media use; DepS: Depressive symptoms; FinS: Family financial status; Gender: female, 0 and male, 1.

**Table 3 ijerph-17-05921-t003:** Summary of model fit indices for all models.

	Fit Indices							
Model	χ^2^	scf	df	*p*	RMSEA	CFI	TLI	SRMR
Model a	67.442	1.123	32	<0.001	0.020	0.976	0.966	0.063
Model b	40.211	1.129	31	0.124	0.011	0.994	0.991	0.047
Model c	92.157	1.117	71	0.047	0.010	0.989	0.984	0.048
Model d	98.859	1.046	71	0.016	0.024	0.977	0.966	0.055

Model a: Autoregressive and cross-lagged paths were constrained to be equal across time. Model b: Autoregression between T5 and T6 on depressive symptoms is set free. Model c: Family financial status and gender are included as covariates.

**Table 4 ijerph-17-05921-t004:** Parameter estimates for the bivariate fixed RICLPMs modelling social media use and depressive symptoms (N = 2891).

Parameter	B	SE	95% CI	*p*	*β*
T1 correlation	0.03	0.02	−0.01 to 0.07	0.095	0.08
Correlated change	0.02	0.01	−0.01 to 0.04	0.233	0.04
Between-person correlation	0.02	0.02	−0.02 to 0.06	0.238	0.07
Cross-lagged paths					
DepS → SoMe	0.13	0.06	0.01 to 0.25	0.039	0.08 to 0.10
DepS T3 → SoMe T5	0.15	0.19	−0.22 to 0.53	0.422	0.12
SoMe → DepS	0.01	0.03	−0.05 to 0.07	0.802	0.01 to 0.01
SoMe T3 → DepS T5	0.14	0.08	−0.03 to 0.31	0.097	0.18
Autoregressive paths					
SoMe → SoMe	0.33	0.05	0.23 to 0.44	<0.001	0.34 to 0.37
SoMe T3 → SoMe T5	−0.08	0.14	−0.35 to 0.19	0.569	−0.08
DepS → DepS	0.12	0.10	−0.08 to 0.31	0.238	0.10 to 0.11
DepS T3 → DepS T5	0.16	0.13	−0.10 to 0.42	0.237	0.17
DepS T5 → DepS T6	0.56	0.07	0.42 to 0.70	<0.001	0.48

Note: B, unstandardized beta weights; β, standardized beta weights; CI, unstandardized confidence intervals; DepS, depressive symptoms; SoMe, social media use. Autoregressive and cross-lagged paths were set equal across time, excluding paths between T3 and T5.

**Table 5 ijerph-17-05921-t005:** Parameter estimates for the bivariate fixed RICLPMs modelling social media use and depressive symptoms with gender and family’s financial status as covariates (N = 2891).

Parameter	B	SE	95% CI	*p*	*β*
T1 correlation	0.03	0.02	−0.00 to 0.07	0.075	0.08
Correlated change	0.01	0.01	−0.01 to 0.04	0.319	0.03
Between-person correlation	−0.00	0.02	−0.04 to 0.03	0.883	−0.01
Cross-lagged paths					
DepS → SoMe	0.12	0.06	−0.01 to 0.23	0.060	0.07 to 0.08
DepS T3 → SoMe T5	0.19	0.19	−0.18 to 0.56	0.325	0.16
SoMe → Deps	0.01	0.03	−0.01 to 0.06	0.898	0.01
SoMe T3 → Deps T5	0.07	0.09	−0.09 to 0.23	0.381	0.11
Autoregressive paths					
SoMe → SoMe	0.31	0.03	0.21 to 0.42	<0.001	0.32 to 0.35
SoMe T3 → SoMe T5	−0.02	0.14	−0.30 to 0.26	0.913	−0.02
DepS → Deps	0.14	0.10	−0.05 to 0.33	0.145	0.12 to 0.14
Deps T3 → DepS T5	0.14	0.13	−0.12 to 0.39	0.291	0.15
DepS T5 → Deps T6	0.53	0.08	0.38 to 0.68	<0.001	0.45

Note: B, unstandardized beta weights; β, standardized beta weights; CI, unstandardized confidence intervals; DepS, depressive symptoms; SoMe, social media use. Autoregressive and cross-lagged paths were set equal across time, excluding paths between T3 and T5.

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
