# Peer review of "Social Media Use and Depressive Symptoms—A Longitudinal Study from Early to Late Adolescence"

_ijerph, 2020, doi:10.3390/ijerph17165921_

Round 1

Reviewer 1 Report

Manuscript ID: ijerph-872637

Social Media Use and Depressive Symptoms –  A Longitudinal Study from Early to Late Adolescence

This paper analyses the within-person effects between active social media use and depressive  symptoms using a five-wave longitudinal dataset of Finnish adolescents. The findings demonstrate that depressive symptoms predicted small increases in active social media use during both early and late adolescence, whereas no evidence of the reverse relationship was found.

The paper is well written and certainly of interests, and the authors have been cautious enough to discuss limitations. However, there is certainly room for clarifications.

Main comments:

  • On page 3, in the section Aims the following sentence needs a bit more discussion: “It is possible that the association between social media use and depressive symptoms is causal, bi-directional, or non-existent”. It would be worth clarifying whether excising research addresses causality issues.

Data:

  • Concern that nearly half only participated once and 26% three or four times. There is a clear attrition issue here that could affect the results especially because the interest is in longitudinal analysis. This requires a bit more discussion.
  • Moreover, I am unclear whether the analysis has been done including anyone, regardless of the times they have been interviewed? If that is the case, some sensitivity checks should be carried restricting the analysis to only those who have been interviewed three times (or four).
  • Also, the authors states that “During 2014–2018 the data collection was conducted in collaboration with schools.” It is not clear if the participation indicated above with 46% being only interviewed once  was the one conducted during 2014–2018. If that is the case, this should and could be clarified.

Models:

  • The models seem to suffer from omitted variable bias, as only family income (as subjective measure) has been included, but there are likely to be other family/household related variables that could affect the dependent variables. This deserves some discussion.
  • The same applied for individual characteristics, in fact only gender seems to be controlled for. Other variables, for example having some siblings, or academic performance (to mention two) could be relevant determinants. Again, this should at least be discussed or referred to excising literature.

Minor comments:

  • Figure 1 is unclear and a bit of explanation would help the reader to understand.

  • Last sentence of page 5 “Missing data were handled using full information maximum likelihood (FIML), which uses all available data to estimate the model without imputing data” talks about missing observations, but it is unclear what has been done with missing observations? Please spell it out.

  • Were individuals observed in each measurement times as reported in table 1 not the same? Please clarify. If that is the case it would be helpful to have a similar table for individuals observed in each measurement times, or at least three or four times.

  • Discussion should be toned down as they authors says “Therefore, it is unclear whether depressive symptoms are a cause or a consequence of social media use. To test the robustness of these associations, we controlled the potential influence of gender and family’s economic status on these associations.” However the analysis offered is interesting and relevant, it does not address endogeneity issues and has omitted variable bias, since many controls are not included as mentioned before, so this should be discussed/ acknowledged.

Author Response

Dear reviewer,

We thank you for the excellent comments, which we found constructive and well justified. We feel that these comments have helped us clarify our argumentation and significantly improve the paper.

Please see the point by point reply attached. 

Best regards,

Authors 

Reviewer 2 Report

Puukko et al.: Social Media Use and Depressive Symptoms –A Longitudinal Study from Early to Late Adolescence

This is a longitudinal study on a large group of Finnish adolescents aimed to examine the correlation between depression and social media use. Authors found that depressive symptoms predicted small increases in active social media use, but no evidence of the reverse relationship was found. The results of this study are strong refutations of that opinion that social media has mainly negative effects on social well-being of adolescents.

The study is well designed, correctly conducted. For the statistical analysis appropriate and sophisticated method was used. Results were moderately interpreted. Strengths and limitations were correctly listed. The manuscript is worth for the attention of the readers of IJERPH.

Minor comment:

In Table 2 two ‘DepS’s and four ‘FinS’s are indicated with different correlation results. The index of these variables might be missing.

Author Response

Dear reviewer,

We thank you for the good comments, which have helped us improve the paper.

Best regards,

Authors 

Reviewer 3 Report

Thank you for the opportunity to review this manuscript. This study examined adolescents’ frequency of social media use and its relation to depressive symptoms. Results did not indicate a significant relationship between these variables, which are not consistent with previous studies indicating a stronger relationship between social media use and depression. Overall, I think this is very well written and presents interesting findings that contribute to the available literature on social media use.

I only have two very minor comments:

Consider adding information regarding the validity of your measures.

There are a couple of small grammatical/formatting errors.

Author Response

Dear reviewer,

We thank you for the comments, which have helped us clarify our argumentation and improve the paper.

Best regards,

Authors 

Reviewer 4 Report

It’s a very difficult issue to discuss the within-person effects between social network emotional expression and depressive symptoms for adolescents from 13-19, because of the close connection with the external influence factors of society, such as parental supervision, freedom of expression, rebellious phase, social habits, and etc. It means, if you want to deeply discuss the adolescents’ depressive symptoms based on the social media, only the frequency factor is far from enough. That’s why the final conclusion of the associations were very small between depressive symptoms and active social media looks like not very convincing, because it is an obvious result. Anyway, maybe proof by contradiction has a certain persuasion, but it’s a more valuable conclusion with what factors will lead depression trend from social media expression. In particular, it is inappropriate to deny that social network to be able to identify depressive tendency. In fact, the index pictures, contents, topics of concern, social events, and some other related information on the social media will help us to know about the psychological state of adolescents. For the paper structure, the materials and methods are clearly described, and the results are throughout. However, I have some questions, whether the culture difference in some way had great influence of the final results? That means, is there any evidence that the proportion of diagnosed depressive disorder less than other countries? How to prove that Finland adolescents have less pressure from the sociality due to its inclusiveness? And another doubt is that whether it is reasonable for only evaluate the frequency, however, is the content matters? For example, whether if the more constantly they chat with friends or family online about their stress or negative emotions, the less loneliness or helpless they will feel, in another word, the lower chance that they might experience depressive symptom.

Author Response

Dear reviewer,

We thank you for the good comments, which we found constructive and well justified.

Altogether, we feel that these comments have helped us clarify our argumentation and significantly improve the paper.

Please see the point by point response attached.

Best regards,

Authors
